# A Novel Surface Descriptor for Automated 3-D Object Recognition and Localization

**DOI:** 10.3390/s19040764

**Published:** 2019-02-13

**Authors:** Liang-Chia Chen, Thanh-Hung Nguyen

**Affiliations:** 1Department of Mechanical Engineering, National Taiwan University, Taipei 10617, Taiwan; 2Department of Mechatronics, School of Mechanical Engineering, Hanoi University of Science and Technology, 1 Dai Co Viet Road, Hanoi 112400, Vietnam; nthung84@gmail.com

**Keywords:** Machine vision, 3-D point cloud, object segmentation, object recognition, object localization, 3-D descriptor

## Abstract

This paper presents a novel approach to the automated recognition and localization of 3-D objects. The proposed approach uses 3-D object segmentation to segment randomly stacked objects in an unstructured point cloud. Each segmented object is then represented by a regional area-based descriptor, which measures the distribution of surface area in the oriented bounding box (OBB) of the segmented object. By comparing the estimated descriptor with the template descriptors stored in the database, the object can be recognized. With this approach, the detected object can be matched with the model using the iterative closest point (ICP) algorithm to detect its 3-D location and orientation. Experiments were performed to verify the feasibility and effectiveness of the approach. With the measured point clouds having a spatial resolution of 1.05 mm, the proposed method can achieve both a mean deviation and standard deviation below half of the spatial resolution.

## 1. Introduction

Nowadays, both 2-D and 3-D machine vision systems are widely integrated with robot manipulators to enhance the flexibility and versatility of modern manufacturing systems. These intelligent integrated systems can accelerate manufacturing and produce efficiently customized products to enhance competitiveness. 2-D machine vision systems [1,2,3,4] are still used in most integrated systems due to their high accuracy and low cost. However, the operation accuracy in these systems is limited by the viewing angle of optical sensing and is especially sensitive to the alignment precision of the jig and fixture employed in locating the workpiece. Systems that involve 3-D data processing can overcome the existing difficulties in 2-D digital imaging by relying on both shape and color information of the objects. These systems have recently been integrated with automated machine manipulators for pick and place applications [5,6,7,8,9,10,11].

One of the most important tasks in 3-D data processing is to determine the position and orientation of the target in 3-D space. This task remains challenging in automation because the target can be of any geometric form and its positioning orientation has six degrees of freedom. Thus, it is difficult to detect the target with an unstructured data input, such as range point clouds. In recent years, several attempts have been made to solve this nontrivial problem. According to their characteristics, the proposed strategies can be divided into graph-based, feature-based, and view-based methods.

Graph-based methods extract the geometric properties of a 3-D shape using a graph, which represents the type and spatial relations between shape components [6,7,12]. In these approaches, the topology graph is built using the detected primitive shapes in the scanned scene. The query graph, which represents the structure of the CAD model and is manually defined by the user, is employed to search the target in the topology graph. Fuchs et al. [6] substituted two parallel lines for a pipe. The distance between two lines is the diameter of the pipe. Skotheim et al. [7] defined a car part as a graph containing two vertices and an edge. One vertex represents a reference plane and the other vertex describes a search plane. The edge of the graph is associated with the distance and relative orientation between the centers of the reference and search planes. More generally, Schnabel et al. [12] presented the scanned data with a topology graph, in which the vertices of the graph are the detected shapes (planes, cylinders, spheres, and cones) and the edges of the graph joined the connected shapes. This technique had been integrated with the service robot for bin picking [10]. Graph-based methods are pose-independent and the graph-based structure can be used for partial matching. However, the matching criteria depend on the structure of the 3-D shape and the computation time involved could be extremely high. Moreover, for graph-based methods, the similarity measure is sensitive to changes in topology. These methods are commonly used in applications, in which the form of the target object is composed of a few simple primitive shapes, such as planes, cylinders, spheres, and cones.

Feature-based methods [13,14,15,16,17,18,19] discriminate 3-D objects by measuring and comparing the geometric and topological properties of 3-D shapes. The points that contain sufficient information to distinguish the shape from others are extracted from the scanned data and defined as feature points. The local information around feature points is then employed to build the local 3-D descriptor. The target object is recognized by matching the scene descriptors with model descriptors. Johnson and Hebert [13] presented the spin image, which was created by projecting the 3-D points into 2-D images for object matching. Chen and Bhanu [14] investigated the relation between the shape index (SI) and the dot product of the surface normal to establish a local surface patch descriptor for object recognition. Zaharescu et al. [15] computed the gradient for each feature point and the histogram of the gradient (HOG) both spatially at a coarse level and a fine level to obtain the MeshHOG descriptor for mesh matching. Drost et al. [16] introduced the point pair feature to describe the geometric information, aligning the model point pairs with the scene point pairs for 3-D object recognition. Rusu et al. [17] considered the relationships between the points in the *k*-neighborhood of the feature point and their estimated surface normals to build the fast point feature histogram (FPFH) descriptor. By encoding the important statistics between the viewpoint and the surface normals on the object into the extended FPFH descriptor to perform the viewpoint feature histogram (VHF), the object and its pose can be simultaneously recognized in the scene point clouds. Salti et al. [18] generated a local histogram according to the distribution of the normal vector at the feature point. These local histograms are then categorized together to form the actual descriptor, which is called the signature of histograms of orientations (SHOT). More recently, geometric information has been used in object recognition [19]. Feature-based maximum-likelihood matching by a simple two-degree-of-freedom analysis of 3D point cloud data demonstrated effect view-based veness in automated object recognition [20]. However, feature-based methods are good when only partial data are available. In general, the descriptors are less discriminative from a global matching point of view. In general, these methods can work best when having geometric details, but are less efficient in comparison to other methods.

View-based methods recognize the target object according to the principle that two 3-D objects are only similar when they look similar from all viewing angles [8,11,21]. In these methods, a real range image acquired from a 3-D scanner can be matched with a set of range images stored in the template database. The database is generally generated from the 3-D model of the object from all possible viewing angles. Liu et al. [8] represented scanned data with a depth-edge map. The database of depth-edge templates is created from a CAD model by detecting the depth discontinuities in the model. Each template is then compared with the depth-edge map to determine the position and location of the target object in the scanned scene. Sansoni et al. [11] separated the scanned data into individual segmented point clouds. Each segmented point cloud is then matched using a commercial software with a set of 3-D templates, which are acquired from different viewpoints of the real object. In the approach proposed by Chen, the segmented object is represented by the curvature-based histogram, which is computed using the shape index value of every point of the segmented object [21]. The target object can be recognized by comparing the computed curvature-based histogram with the template histograms in the database. View-based methods can be applied to almost all kinds of objects. However, the disadvantage is that the comparison of the database and the acquired data is time-consuming.

From the above review, it can be seen that one of the reasons why the 3-D vision system has not been widely used in robotic automation is the long computation time required for practical application. In view of this, a fast and effective method for 3-D object recognition and localization is proposed to deal with the difficulties existing in this task. The main idea of the approach is that two objects are similar if the distributions of their surface areas in their oriented bounding boxes (OBBs) are the same.

In this study, the proposed approach for 3D object recognition and localization can deal with the difficulties existing for practical applications such as robotic automation. The developed method proposes a new index for 3-D object recognition with a good efficiency in operation and robustness to illumination variations. In addition, the key solution can recognize stacked objects with arbitrary orientation. The main idea of the approach starts from the point that two objects are similar if the distributions of their surface areas in their OBBs are the same. The method comprises two main stages. In the first stage, the proposed regional area-based descriptors are computed for implementing the shape-matching algorithm for reliable object recognition. In the second stage, the position and orientation of the target are initially determined by aligning the OBBs and further refinement by the iterative closest point (ICP) algorithm.

The rest of this paper is organized as follows. Section 2 presents the proposed method for object recognition and localization employing regional area-based descriptors. The experimental results and analysis are shown in Section 3. Section 4 discusses the characteristics and limitations of the developed method in detail. Finally, the conclusions and further work are summarized in Section 5.

## 2. Methodology

Given a point cloud that represents the scene of the randomly stacked objects in an unstructured bin, the task is to recognize and localize the target object in the scene point clouds. To deal with the task, the proposed approach first separates the scene point clouds into individual object point clouds [22]. Each segmented object can be represented by a feature vector, which is computed according to its OBB and object surface area. The feature vector is then matched with the feature vectors kept in the database, which represent the different views of the object model. According to the matching results, the transformation matrix can be initially computed to align the segmented object with the object model. Finally, the 3-D position and orientation of the target object in the scene point clouds can be estimated using the ICP algorithm. The overview and flowchart of the proposed method are shown in Figure 1 and Figure 2, respectively. In addition, the operation procedure of the proposed method is described in Algorithm 1.


**Algorithm 1:**
Input: Measured point cloud *O* and model point cloud *M.*Output: Position and Orientation of objects.
***Segmentation***: Segment scene point cloud into individual object point clouds (Segmentation) *O* = {*O_j_*, *j* = 0, 1, …, *n*}. (Section 2.1)Compute descriptor of each segmented individual object point cloud *FV_O_* = {*FV_Oj_*, *j =* 0, 1, …, *n*} (Section 2.2)***3D Virtual Camera***: Extract m point clouds corresponding to m different views of the model point cloud using 3D virtual sensor (result: *M* = {*M_i_*, *i =* 0, 1, …, *m*}) and compute the descriptors (result: *FV_M_* = {*FV_Mi_*, *i =* 0, 1, …, *m*}). (Section 2.3)***Recognition***: Match the descriptors of *O_j_* and *M*, determine the best matching, and obtain corresponding points of *O_j_* in *M, M_Oj_*. (Section 2.4.1 and Section 2.4.2)***Localization***: Align the OBB of the point cloud *O_j_* and OBB of the corresponding point cloud *M_Oj_*. Obtain the initial transformation matrix, and then apply the Iterative Closest Point (ICP) algorithm for refinement. (Section 2.4.3)


### 2.1. Object Segmentation

The point cloud data acquired from the 3-D scanner comprises information of one view of different objects in the scene. In order to obtain the segmented point clouds corresponding to one view of an object for a later recognition task, the scene point clouds should be further separated into smaller point clouds. In recent years, many techniques have been developed to obtain the individual parts from the point cloud data. The popular techniques are the region growing method [23,24], *k*-nearest neighbors clustering algorithm [25,26], and graph theoretic approach [27,28,29].

In this study, the developed 3-D object segmentation algorithm for randomly stacked objects [29,30] is employed to obtain the segmented point clouds corresponding to the object in the scene point clouds. The basic idea of the proposed algorithm is that the points located geometrically farther from the surface boundary of the projected object are more likely to belong to the same object than the other points (as shown in Figure 3b). The points far away from the surface boundary can be used as an internal seed marker for object region growth (as illustrated in Figure 3c). With a flooding algorithm, the scene point clouds can be clearly separated into smaller individual object point clouds (see Figure 3d). The general process of the proposed algorithm is presented in Figure 3.

### 2.2. Regional Area-based Descriptor

To find the similarity between the point cloud model *M* and the segmented point cloud *O*, which is extracted using the object segmentation algorithm, the feature descriptor of the segmented object is compared with the feature descriptors of different views of the model. A feature descriptor is normally defined by two essential elements [30], namely the OBB and surface area of the object. An OBB consists of a corner and three principle vectors (shown in Figure 4). The surface area of the object defines the histogram of all the surface areas within the OBB. In general, the total number of subdivided boxes is *k*_1_ × *k*_2_ × *k*_3_ when the OBB is defined by *k*_1_, *k*_2_, and *k*_3_ (shown in Figure 5). The surface area in each subdivided box (*V_ijk_*) is described as *S_v_*, in which *v* = *kk*_1_*k*_2_ + *jk*_1_ + *i*. Let *S* be the total surface area of the segmented object in the OBB and *f_v_* equal *S_v_*/*S*. The feature descriptor can be described as follows:(1)$$FV=\{C,CC_{1},CC_{2},CC_{3},f_{0},\ldots ,f_{v},\ldots ,f_{n_{V}}\},$$
where
*n_V_* = *k*_1_*k*_2_*k*_3_ − 1;***C***: corner vector;***CC***_1_, ***CC***_2_, and ***CC***_3_: principle vectors corresponding to the maximum, middle, and minimum dimensions of the OBB, respectively.

Finally, according to *S_v_*, the regional area-based descriptor can be built as shown in Figure 6. In this example, the OBB of the object point clouds is subdivided into eight sub-boxes. In addition, three parameters *k*_1_, *k*_2_, and *k*_3_ are set to equal 2.

The proposed descriptor utilizes the distribution of the object’s surface area inside the OBB of the object to represent the object in 3D space. The OBB regional area-based descriptor is invariable to arbitrary poses of the objects, because the surface area is an intrinsic property that is completely independent of object positions and orientations in space. Furthermore, the developed approach is robust to numerous variations of surface sampling density and noise generated from the measurement process. In order to match the model point cloud and the segmented point cloud extracted using the object segmentation algorithm, the feature descriptor of the segmented object is compared with the feature descriptors of various views of the model in the database using normalized cross-correlation (NCC).

The total number (*n*) of subdivided boxes in the OBB is an important parameter for computation efficiency. When the OBB contains more subdivided boxes, matching normally takes more time and the matching accuracy is thus increased. To shorten the matching procedure, fewer subdivided boxes are preferable. However, to ensure matching accuracy, the number of subdivided boxes should be adequately set to achieve meaningful matching. Figure 7 illustrates three examples of the regional area-based descriptors of an L-shape object with three different types of segments, in which *n* (*= k*_1_
*×*
*k*_2_
*×*
*k*_3_) is computed as 5 × 5 × 1, 5 × 4 × 3, and 5 × 5 × 4, respectively, with the running time being 0.15, 0.35, and 0.65 s, respectively.

#### 2.2.1. Estimation of Oriented Bounding Box

The OBB of an object is a rectangular bounding box that covers all object point clouds. The orientation of an OBB can be determined using the covariance matrix [31]. The associated algorithm is described in the following steps:

Calculate the center $\bar{p}$ of the point cloud *P*;

Compute the covariance matrix:(2)$$COV=\frac{1}{n}\sum_{i=1}^{n}\left(p_{i}-\bar{p}\right){\left(p_{i}-\bar{p}\right)}^{T};$$

Extract the eigenvectors {***v***_1_, ***v***_2_, ***v***_3_} from the covariance matrix;

Determine the dimensions of the object defined in each eigenvector using the distances between the nearest and farthest projected points.

The corner **C**(x_C_, y_C_, z_C_) farthest from the center $\bar{p}$ will be chosen from eight corners of the OBB. Then, the vectors **CC**_i=1–3_ can be established from the largest, medium, and smallest dimensions of the OBB, respectively. However, the distances between the corners and the center point are sometimes indistinguishable or the dimensions of the OBB along **CC**_i=1–3_ are similar. In these circumstances, all possibilities have to be taken into account to determine the best choice using NCC mentioned in Section 2.4.2.

#### 2.2.2. Simplified Regional Area-based Descriptor

The normalized surface area in each subdivided box can be calculated as follows:(3)$$f_{v}=\frac{S_{v}}{S}\approx \frac{\sum_{i=1}^{m}S_{i}}{S}\approx \frac{n_{s}}{n_{t}},$$
where *n_t_* is the number of triangles in the triangle mesh and *n_s_* is the number of triangles that are inside the subdivided box.

In order to reduce the computation time for the regional area-based descriptor, the number of triangles inside the subdivided box can be replaced by the number of points inside the subdivided box. In addition, the number of triangles in the triangle mesh can be substituted by the number of points in the point clouds, as expressed below:(4)$$f_{v}=\frac{S_{v}}{S}\approx \frac{n_{v}^{p}}{n}=f_{v}^{p},$$
where $n_{v}^{p}$ is the number of points inside the *v*^th^ subdivided box and *n* is the number of points in the object point clouds.

By applying Equation (4), a new form of the regional area-based descriptor can be expressed as follows:(5)$$FV=\{C,CC_{1},CC_{2},CC_{3},f_{0}^{p},\ldots ,f_{v}^{p},\ldots ,f_{n_{V}}^{p}\}.$$

Equation (5) is the simplified regional area-based descriptor. The descriptor captures less precise information included in the surface area-based descriptor, but it still retains most of the discriminative power of the surface area-based descriptor. The regional area-based descriptor and its simplification are shown in Figure 8b. As can be seen, the distribution of the surface area inside the OBB of the object in this case is almost the same as the distribution of points inside the OBB. Furthermore, the efficiency of simplification is demonstrated by six tests in Figure 9.

### 2.3. D Virtual Camera

A virtual sensor is developed to extract the template point clouds corresponding to various views of the model. The virtual sensor with the same internal properties as the real one is located at the origin of the world coordinate system. The central axis of the virtual camera is defined as the *z*-axis. The CAD model of the object is positioned on the *z*-axis. The working distance, *t_z_,* is shown in Figure 10. The reference point clouds are generated by rotating the model around the *x*-axis with every increment *θ_x_* and around the *y*-axis with every increment *θ_y_*. For each rotation, a set of point clouds is created corresponding to one view of the model. Figure 11 shows the different views of the CAD model in Figure 10.

### 2.4. Feature Matching

Given a feature vector ***FV****_O_* that represents the object point clouds *O*, the matched feature vector ***FV****_M_* that represents a view of the model needs to be estimated. The feature vector ***FV****_M_* is built through the database generation process in the offline phase. The feature vector ***FV*** comprises the OBB parameters and the regional area-based descriptor. In the first step, the OBB parameters between the object point clouds and the object model are compared for best matching from a series of viewpoints [30]. When the OBB matching result satisfies the preset condition, the regional area-based descriptor between the object point clouds is then matched with the object model defined from the viewpoint. The general process for matching the object point clouds with the model is shown in Figure 12.

#### 2.4.1. OBB Matching

The OBB matching process determines the views of the object model comprising the same dimensions as the object point clouds. The OBB of each template point cloud is represented by three vectors, including ***CC****_M_*_1_(*x_Mmax_*, *y_Mmax_*, *z_Mmax_*), ***CC****_M_*_2_(*x_Mmid_*, *y_Mmid_*, *z_Mmid_*), and ***CC****_M_*_3_(*x_Mmin_*, *y_Mmin_*, *z_Mmin_*), while the OBB of the object point clouds is denoted by ***CC***_1_(*x_max_*, *y_max_*, *z_max_*), ***CC***_2_(*x_mid_*, *y_mid_*, *z_mid_*), and ***CC***_3_(*x_min_*, *y_min_*, *z_min_*), respectively. These two OBBs should be satisfied by the following equation:(6)$$d_{corr}=\frac{1}{3}\left(\frac{\left\|CC_{M1}-CC_{1}\right\|}{\left\|CC_{M1}\right\|}+\frac{\left\|CC_{M2}-CC_{2}\right\|}{\left\|CC_{M2}\right\|}+\frac{\left\|CC_{M3}-CC_{3}\right\|}{\left\|CC_{M3}\right\|}\right)<d_{thresh},$$
where *d_thresh_* is the given adequate threshold.

#### 2.4.2. Matching Criteria for Regional Area-based Descriptor

The regional area-based descriptor presents the histogram of the surface area of the object. The resemblance between the object and model descriptors is measured using NCC. Let *F_O_* = {*f_v_*, *v* = 0, …, *n_V_*} be the regional area-based descriptor of the object point clouds and *F_M_* = {*f_Mv_*, *v* = 0, …, *n_V_*} be the regional area-based descriptor of the template point clouds. The NCC between *F_O_* and *F_M_* is computed as follows:(7)$$C\left(F_{O},F_{M}\right)=\frac{\sum_{v=0}^{n_{V}}\left(f_{v}-\bar{f}\right)\left(f_{Mv}-{\bar{f}}_{M}\right)}{\sqrt{\sum_{v=0}^{n_{V}}{\left(f_{v}-\bar{f}\right)}^{2}\cdot \sum_{v=0}^{n_{V}}{\left(f_{Mv}-{\bar{f}}_{M}\right)}^{2}}},$$
where $\bar{f}=\frac{1}{n_{V}+1}\sum_{v=0}^{n_{V}}f_{v}$ and ${\bar{f}}_{M}=\frac{1}{n_{V}+1}\sum_{v=0}^{n_{V}}f_{M_{v}}$.

If the coefficient *C*(*F_O_*, *F_M_*) is larger than a given threshold, the matching result is good and the two feature vectors ***FV****_O_* and ***FV****_M_* can be adopted to estimate the initial transformation matrix between the object point clouds and the model.

#### 2.4.3. Transformation Estimation and Refinement

Through the matching step, the correspondence feature vectors ***FV****_O_* and ***FV****_M_* are obtained. According to these feature vectors, the initial transformation matrix ***T****_initial_* between the object point clouds and the model can be estimated by aligning the frame {***C****_M_*, ***v****_M_*_1_, ***v****_M_*_2_, ***v****_M_*_3_} that represents the model to the frame {***C***, ***v***_1_, ***v***_2_, ***v***_3_} that represents the object point clouds.
(8)$$\left[\begin{array}{cccc} x_{C} & x_{C}+v_{11} & x_{C}+v_{21} & x_{C}+v_{31}\\ y_{C} & y_{C}+v_{12} & y_{C}+v_{22} & y_{C}+v_{32}\\ z_{C} & z_{C}+v_{13} & z_{C}+v_{23} & z_{C}+v_{33}\\ 1 & 1 & 1 & 1 \end{array}\right]=T_{initial}\left[\begin{array}{cccc} x_{MC} & x_{MC}+v_{M11} & x_{MC}+v_{M21} & x_{MC}+v_{M31}\\ y_{MC} & y_{MC}+v_{M12} & y_{MC}+v_{M22} & y_{MC}+v_{M32}\\ z_{MC} & z_{MC}+v_{M13} & z_{MC}+v_{M23} & z_{MC}+v_{M33}\\ 1 & 1 & 1 & 1 \end{array}\right]$$

The accuracy of initial pose estimation is limited by the set of templates in the database. Each template describes only one view of the object and the number of templates is restricted; hence, the matched template may not be exactly the same as the detected object. For example, some small parts of the detected object are missed from the template. Hence, the estimated 3-D pose may be incorrect. Additionally, the measurement data can lose some high-frequency information (such as the edges of the object) owing to limitation of the measurement system. The measurement errors can reduce the accuracy of the initial 3-D pose estimation. Therefore, it is necessary to refine the estimated initial 3-D pose. The algorithm often performed to obtain the refined transformation is the ICP algorithm [32,33]. The ICP algorithm is a matching process that minimizes the fitting deviation between two matching point clouds, and iteratively refines the transformation through minimizing the distance between the points of the object point clouds and the model. The steps of the ICP algorithm are described as follows:For each point ***p*** ∈ *O*, find the closest point ***q*** ∈ *M*;Estimate the rotation matrix ***R*** and translation vector ***t*** that minimize the root mean squared distance;Transform *O_k_*_+1_ ← *Q*(*O_k_*) using the estimated parameters;Terminate the iteration when the change in error falls below the preset threshold.

## 3. Experimental Results and Analysis

The feasibility of the proposed methodology is verified by comparing simulated and actual data obtained from experiments on industrial objects. Figure 13 shows the experimental setup for the robot pick and place application that involves the regional area-based descriptor. The developed 3-D scanner has been integrated with the robotic arm to acquire the 3-D point clouds that represent the randomly stacked objects in the scene. The measurement volume of the 3-D scanner is approximately 147 × 110 × 80 mm^3^. The simulation data provided by Industrial Technology Research Institute (ITRI) comprise the database of six different work parts. The dimensions of each object in the ITRI database are shown in Table 1. The resolution of the scene point clouds in the database is greater than 0.5 mm. Actual data are measured using the developed 3-D optical scanner according to the random speckle pattern projection principle and the triangulation theory. The depth resolution of the measured data is about 0.3 mm and the spatial resolution is 1.05 mm. In the experiment, the datasets are processed on a computer with a Core i5 processor (3.40 GHz and 4 GB RAM).

A viewpoint is defined as a set of six parameters, with three position parameters (*x, y, z*) defining the spatial position of the 3-D sensor, and three orientation parameters (R_x_, R_y_, R_z_) defining the direction of the sensor. The accuracy of a point cloud in the measurement using the 3-D sensor depends on the angle of incidence of the sensor on the surface. The ideal angle is π/2; that is, the closer the angle of incidence of the sensor is to the normal surface direction, the more accurate the measured points. To ensure quality of the measured data, a quality criterion is included in the experimental system. This criterion states that acquired point clouds detected using local normal vectors should satisfy a uniform distribution condition. Therefore, the simplified regional area-based descriptor is utilized for all tested data.

### 3.1. Case Study on Simulated Data

In testing the simulation data, Gaussian random noise is added to the object point clouds with increasing standard deviation on *σ* from 0.001 to 1.0. Following this, the object descriptor is compared with the descriptors of the templates to determine the correlation coefficient using Equation (7). For each model, the correlation coefficient corresponding to each Gaussian noise level is the average of 30 values, which measure the similarity between 30 different object point clouds with the model. The accuracy of the proposed method is evaluated by determining the root mean squares (RMS), translation, and rotation errors. In addition, the computation time of the object recognition and localization process is measured to evaluate the efficiency of the proposed method.

Figure 14 shows examples of input object point clouds with different Gaussian noise levels. The effect of Gaussian noise on matching between the object descriptor and model descriptor is illustrated in Figure 15. In this experiment, the correlation coefficients obtained can exceed 0.8 with a noise level of σ below 1.0.

The accuracy of the 3-D object recognition and localization algorithm can be evaluated by RMS, translation, and rotation errors. The RMS error is calculated using the distance between corresponding points in the CAD model (***p****_itrue_*) and transformed point clouds (***p****_ialg_*) under the estimated transformation matrix. Translation error (*T_err_*) is the absolute difference between the true and computed translation vectors (***T****_true_* and ***T****_alg_*). Rotation error (*q_err_*) is the absolute difference between the true and computed unit quaternions (***q****_true_* and ***q****_alg_*) representing the rotations of objects in 3-D space.
(9)$$T_{err}=\left\|T_{alg}-T_{true}\right\|,$$
(10)$$q_{err}=\left\|q_{alg}-q_{true}\right\|,$$
(11)$$RMS_{err}=\sqrt{\frac{\sum_{i=1}^{n}{\left(p_{ialg}-p_{itrue}\right)}^{2}}{n}}.$$

Figure 16, Figure 17 and Figure 18 show the RMS, translation, and rotation errors, respectively, of six different types of objects in the ITRI *(Industrial Technology Research Institute)* database. The maximum RMS error is smaller than 0.15 mm. The translation errors do not exceed 0.06 mm. In addition, the rotation errors are smaller than 0.005°.

To evaluate the efficiency of the proposed recognition and localization algorithm, the simplified regional area-based descriptor is employed. This descriptor captures less precise information included in the surface area-based descriptor, but it retains most of the discriminative power of the regional area-based descriptor. The computation time of the object recognition and localization tasks is summarized in Figure 19. The average time required to recognize and localize the object is less than 0.3 s for all tested objects.

The comparison of performances between the proposed approach and existing methods is reported in Table 2. As can be seen, the developed method outperforms in pose estimation accuracy and computation cost. In terms of computational efficiency, it is worth noting that the benefits of speed offered by the proposed method outweigh the others. Thus, regional area-based object recognition and the localization algorithm can achieve a real-time and accurate pose estimation of 3D objects in cluttered range images.

### 3.2. Case Study on Measured Data

In the experiments with the measured data, different types of objects were selected to test the effectiveness of the proposed algorithm. The samples are randomly stacked on the table to ensure randomness in their positions and orientations. The performance of object matching can be evaluated by judging the distance from each point in the measured cloud to the closest point in the model [34]. Denote *d_i_* as the distance between a point (***p****_i_*) in the measured cloud and its closest point (***q****_i_*) in the model (shown in Figure 20). Then, the mean distance, *μ,* and standard deviation, *σ*, can be computed to evaluate the matching condition as follows:(12)$$\mu =\frac{1}{n}\sum_{i=1}^{n}d_{i},$$
(13)$$\sigma =\sqrt{\frac{\sum_{i=1}^{n}{\left(d_{i}-\mu \right)}^{2}}{n-1}},$$
where *n* is the number of points in the measured point clouds.

As seen in Figure 21, the location and orientation of all objects can be effectively detected by using the developed method. Three experimental cases are shown, in which three, seven, and two various parts are contained in each case, respectively, for object recognition and localization. The correlation coefficients in the matching process are higher than 0.9. For all tested data, the localization of each part can be completed within 0.5 s. In addition, the mean deviation ranges between 0.180 mm and 0.469 mm and the standard deviation ranges between 0.168 mm and 0.484 mm (shown in Table 3).

In Figure 22, two cases were further tested for the verification of handling more complexity. Case 1 has five parts with an objective to detect a connector, while Case 2 has five parts with an objective to detect a 3D-printed hammer. By using the developed method, two connectors were effectively detected with part orientation determined in Case 1. In Case 2, the 3D-printing hammer, which is partly overlapped by a toy model, can also be detected effectively with its orientation localized.

## 4. Discussion

The proposed methods have been implemented and tested on non-overlapping, multiple overlapping, and stacked objects in 3-D scene point clouds. In this section, the advantages and limitations of object recognition and localization are discussed. The developed method represents the object in 3-D space based on the distribution of the object’s surface area inside the OBB. Surface area is an intrinsic surface property and independent of surface sampling; hence, the developed feature descriptor is invariant to arbitrary rotations and translations of the object. In addition, the feature descriptor is less sensitive to surface sampling and noise. As illustrated in Figure 15, the correlation coefficients exceed 0.8 with a noise level σ equal to 1.0.

The effectiveness and accuracy of the proposed 3-D object recognition and localization have been tested on both simulated and measured data. The accuracy of the proposed method in the experiment with simulated data is evaluated in terms of RMS, translation, and rotation errors, which were found to be below 0.15 mm, 0.06 mm, and 0.005°, respectively.

The proposed algorithm employs a point-to-point ICP algorithm to match the scene point clouds with the model; hence, the accuracy can be affected by the resolution of the scene point clouds, which is about 0.5 mm in the experiment. Therefore, some important features of the object located in the small surface regions are only represented by a small number of measured points, thus causing errors when estimating the position and orientation of the target object. For example, the scene nos. 19 and 28 of the finefood object shown in Figure 23, respectively, are only represented by a rather small number of point clouds. Due to this reason, the RMS and rotation errors of scene nos. 19 and 28 of the finefood object shown in Figure 16 (in Page 15) and Figure 18 (in Page 16), respectively, are larger than those of the other scenes.

Errors in the initial transformation estimation process depend considerably on the rotation increments *θ_x_* and *θ_y_* of templates generated by the virtual camera. To enhance the matching accuracy, the rotation increment is controlled to be smaller. However, a smaller increment can easily generate a significant increase in the runtime of the operation process. Take for instance an experiment having 10,000 points, such as Wrench shown in Table 2; its computation time in the case *θ_x_* = *θ_x_* = 0.157 (rad) was 3 s, which is much longer than that in the case *θ_x_* = *θ_x_* = 0.785 0.157 (rad), at merely 0.15 s.

In the experiment with the actual measurement data, the accuracy of the proposed method is estimated according to the matching deviation between the object point clouds and the model. The performance of object matching is evaluated using the mean distance and its standard deviation. With the measured point clouds having a depth resolution of 0.3 mm and spatial resolution of 1.05 × 1.05 mm^2^, the proposed method can achieve a mean deviation below 0.47 mm and standard deviation below 0.49 mm. To increase the accuracy of the developed algorithm, the best way is to improve the quality and resolution of the measured point clouds.

The efficiency of the 3-D object recognition and localization algorithm is a very important parameter for the 3-D vision system in practical applications. In the experiment, for a set of more than 16k points in one 3-D image map, the proposed method requires only 0.5 s for each recognized object using a common PC, and it can be reduced to 0.1 s by simplification. To substitute it for a regional area-based descriptor, the measured point cloud is required to have a certain level of density and uniform distribution. The computation cost of the algorithm is proved to be efficient for in-line industrial automation.

## 5. Conclusions

In this study, a new method for automated 3-D object recognition and localization has been developed using 3-D point clouds. Experimental results indicate the effectiveness of the proposed method. In addition, the developed approach can accurately detect the position and orientation of the target objects, which are randomly stacked in an unstructured bin. The detection accuracy is affected by the spatial and depth resolution of the measured point cloud. The proposed method can achieve a localization accuracy better than half of the spatial resolution of the point cloud. The developed algorithm would be particularly useful for the automation of workpiece manipulation and handling in manufacturing sectors. Enhancement of computational efficiency of the object recognition process is achievable by further employing parallel computing techniques.

## Figures and Tables

**Figure 1 sensors-19-00764-f001:**
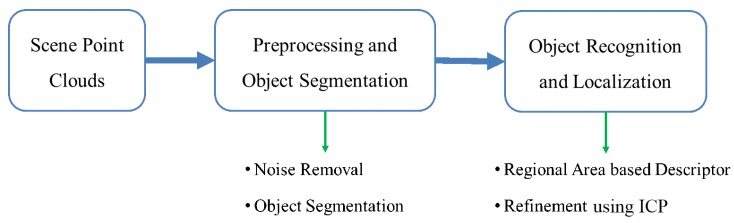
Overview of 3-D object recognition and localization.

**Figure 2 sensors-19-00764-f002:**
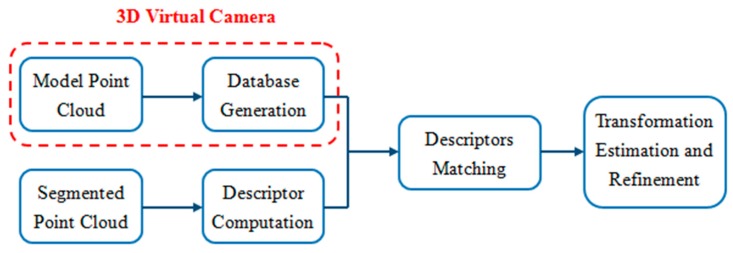
Flowchart of the proposed 3D object recognition and localization algorithm.

**Figure 3 sensors-19-00764-f003:**
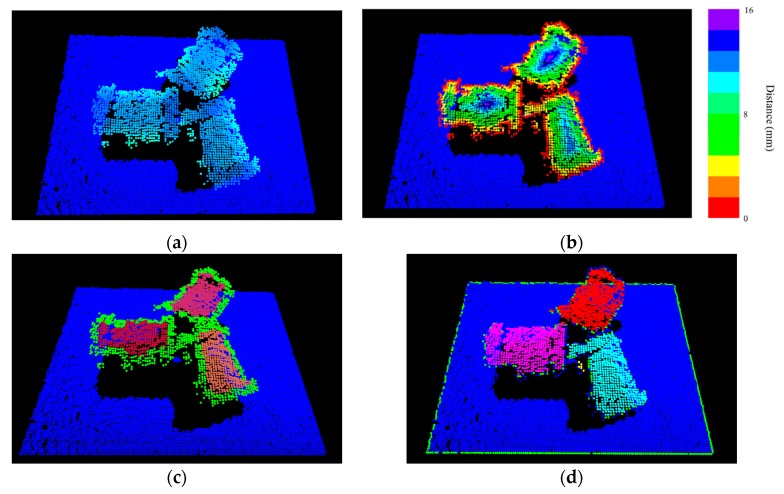
Principle of the proposed 3-D object segmentation method. (**a**) Measured point clouds. (**b**) Distance map. (**c**) Generated markers. (**d**) Segmentation result of 3-D point clouds.

**Figure 4 sensors-19-00764-f004:**
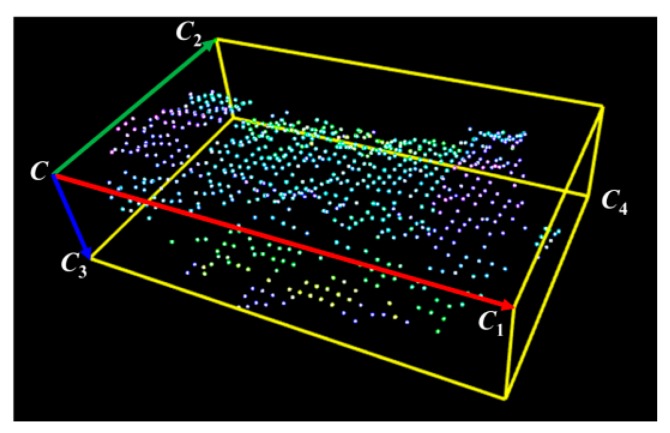
Oriented bounding box of segmented point clouds.

**Figure 5 sensors-19-00764-f005:**
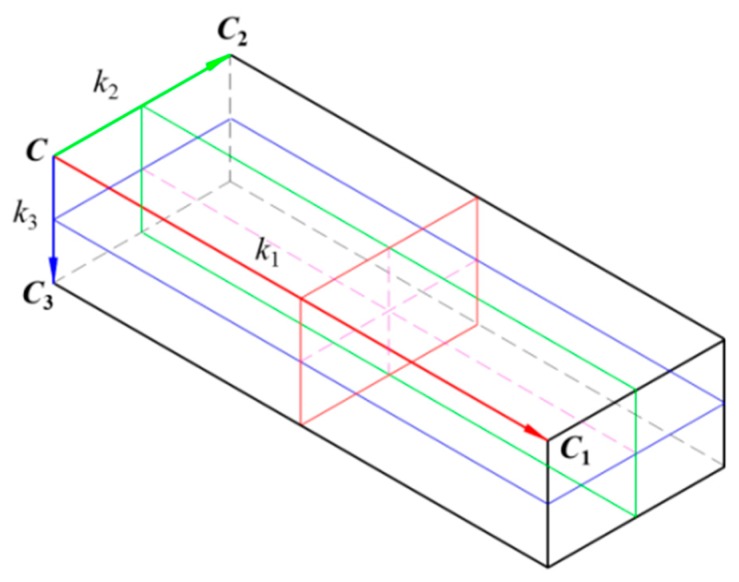
OBB of object subdivided into n (= *k_1_* × *k_2_* × *k*_3_) sub-boxes.

**Figure 6 sensors-19-00764-f006:**
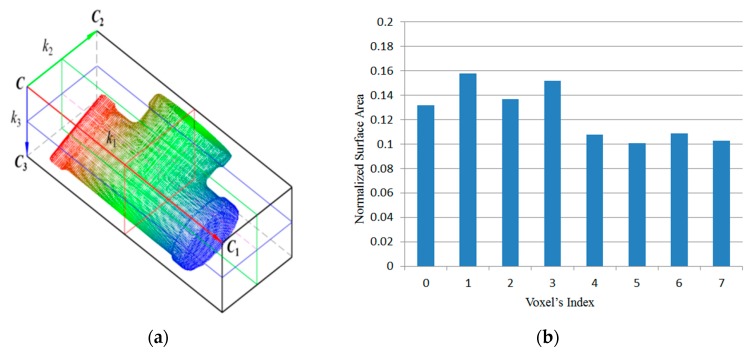
Regional area-based descriptor. (**a**) Object point clouds. (**b**) Regional area-based descriptor of the object.

**Figure 7 sensors-19-00764-f007:**
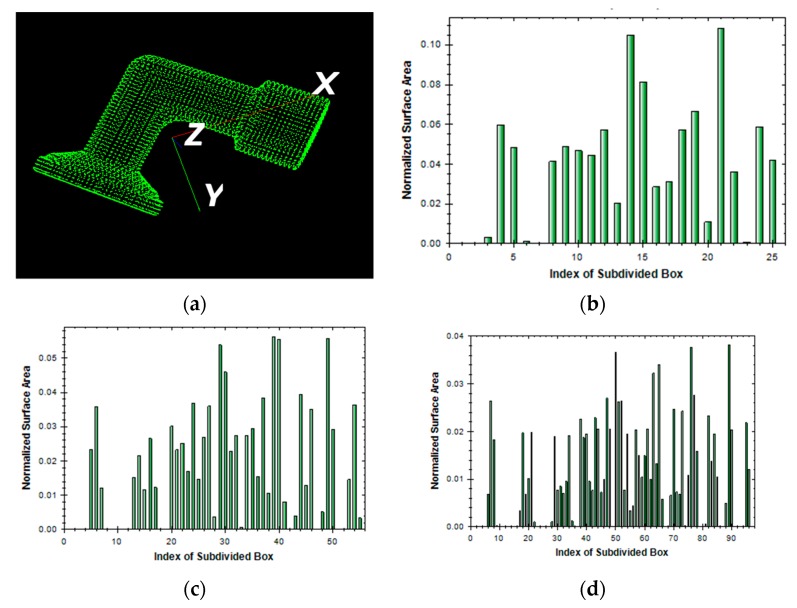
An example of regional area-based descriptor of an L-shape object with different number of subdivided boxes in OBB n = *k*_1_ × *k*_2_ × *k*_3_ (**a**) point cloud; (**b**) *k*_1_ × *k*_2_ × *k*_3_ = 5 × 5 × 1; (**c**) *k*_1_ × *k*_2_ × *k*_3_ = 5 × 4 × 3; (**d**) *k*_1_ × *k*_2_ × *k*_3_ = 5 × 5 × 4.

**Figure 8 sensors-19-00764-f008:**
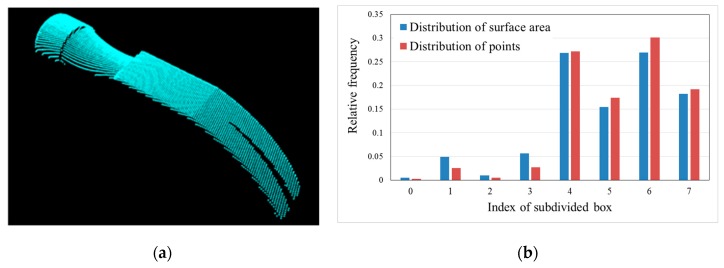
Distribution of surface area and points inside the OBB of the object. (**a**) Object point clouds. (**b**) Regional area-based descriptor of the object (blue) and its simplification (red).

**Figure 9 sensors-19-00764-f009:**
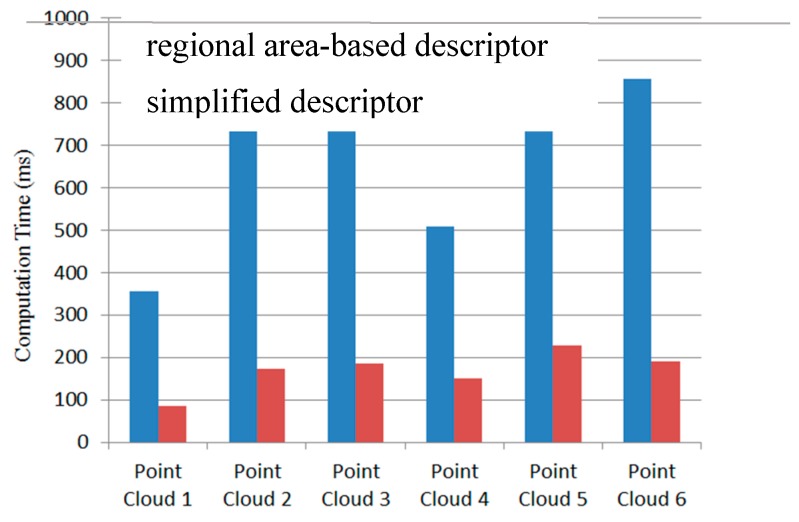
Comparison of computation time between original regional area-based descriptor and simplified descriptor inside OBBs of six point clouds.

**Figure 10 sensors-19-00764-f010:**
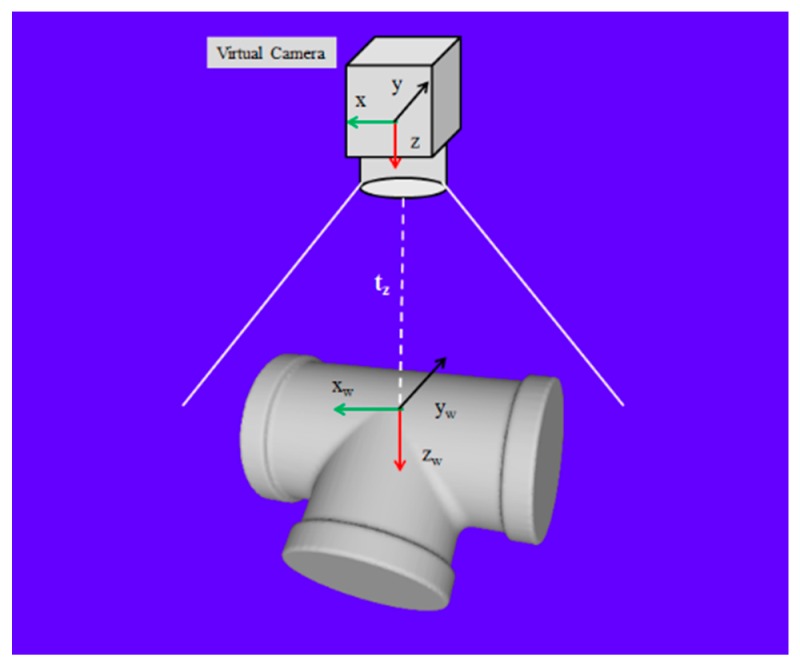
Virtual camera and model.

**Figure 11 sensors-19-00764-f011:**
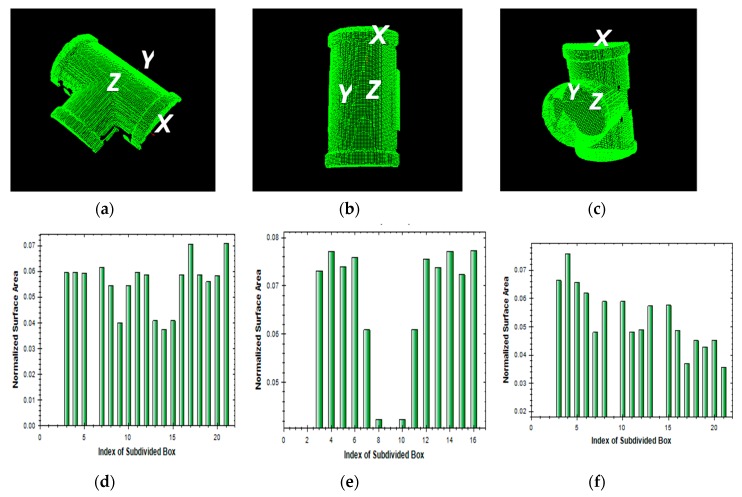
3D virtual sensor. (**a**–**c**) Point clouds corresponding to various views of the model; (**d**–**f**) regional area-based descriptors of point clouds of (**a**–**c**) cases, respectively.

**Figure 12 sensors-19-00764-f012:**
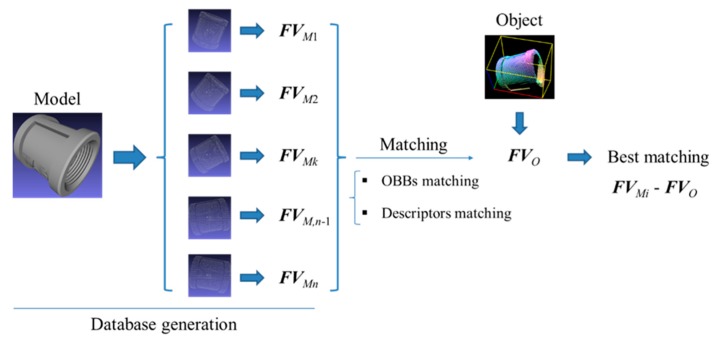
Flowchart for matching object point clouds with the model.

**Figure 13 sensors-19-00764-f013:**
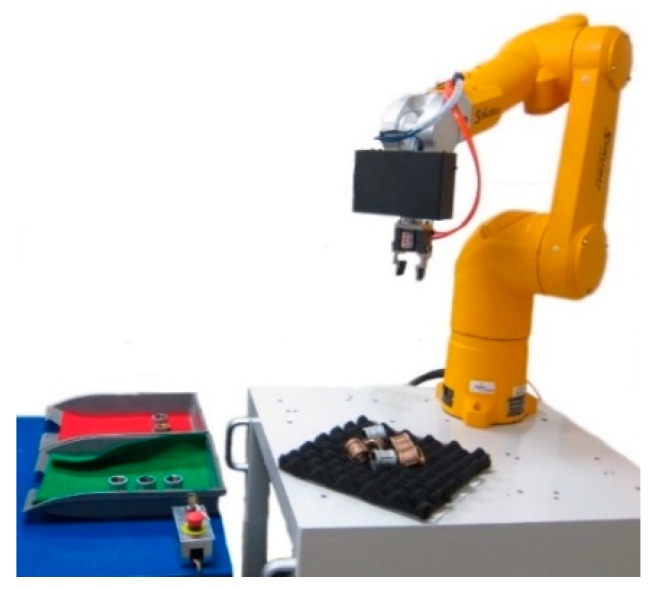
System setup for pick and place application using regional area-based descriptor.

**Figure 14 sensors-19-00764-f014:**
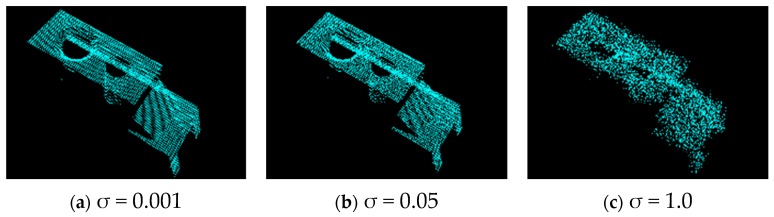
Object point clouds with Gaussian noise added and standard deviation of 0.001, 0.05, and 1.0.

**Figure 15 sensors-19-00764-f015:**
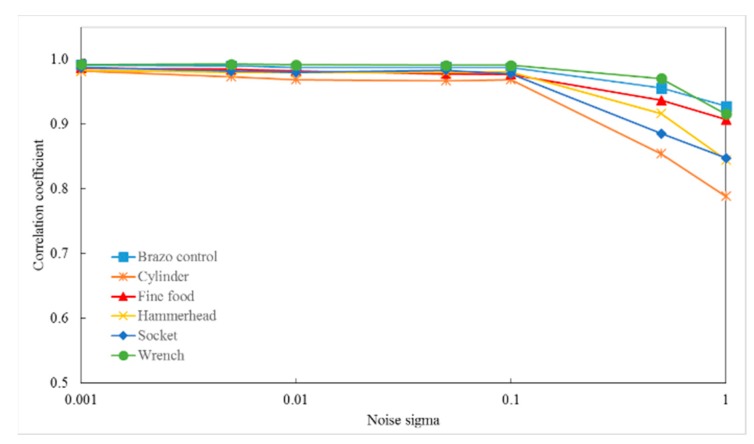
Effect of Gaussian noise on matching between object descriptor and model descriptor.

**Figure 16 sensors-19-00764-f016:**
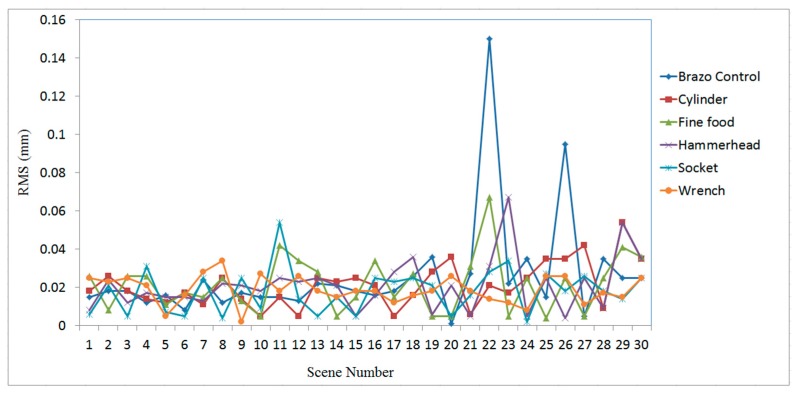
RMS errors obtained using the proposed algorithm for different types of objects in ITRI database.

**Figure 17 sensors-19-00764-f017:**
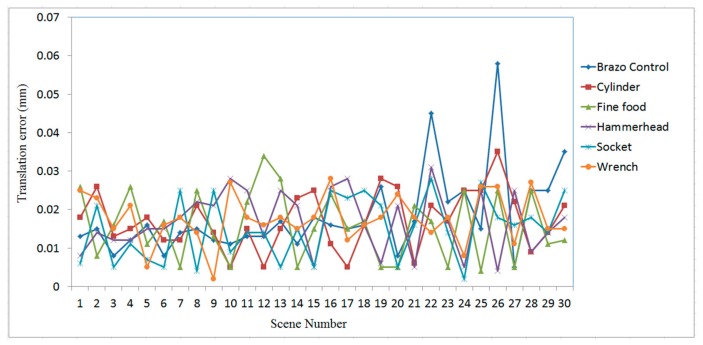
Translation errors obtained using the proposed algorithm for different types of objects in ITRI database.

**Figure 18 sensors-19-00764-f018:**
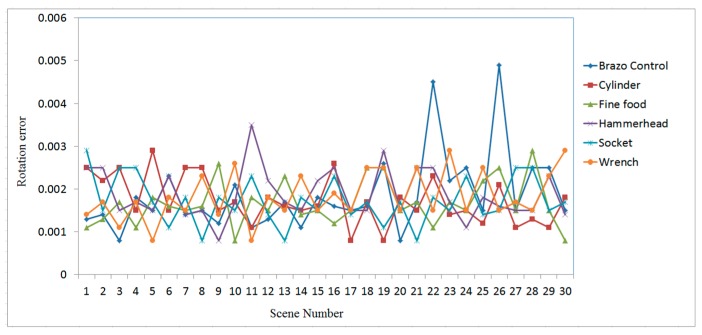
Rotation errors obtained using the proposed algorithm for different types of objects in ITRI database.

**Figure 19 sensors-19-00764-f019:**
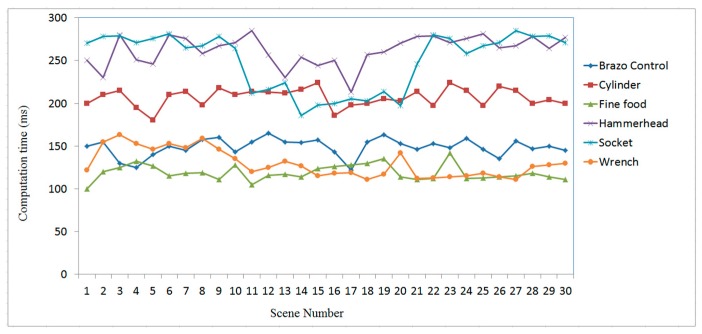
Time consumed for 3-D object recognition and localization using the proposed algorithm.

**Figure 20 sensors-19-00764-f020:**
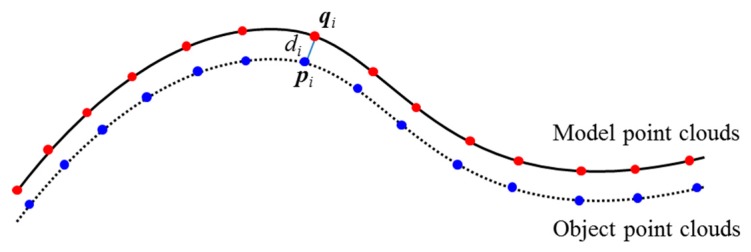
Query point **p**_i_ in measured point clouds and its nearest point, ***q***_i_, in the model.

**Figure 21 sensors-19-00764-f021:**
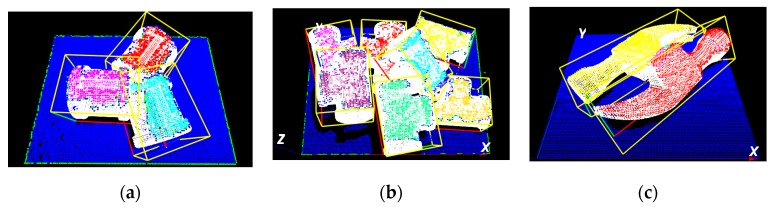
Object recognition and localization results for different types of objects: (**a**) three parts randomly stacked; (**b**) seven parts randomly stacked and (**c**) two parts randomly stacked.

**Figure 22 sensors-19-00764-f022:**
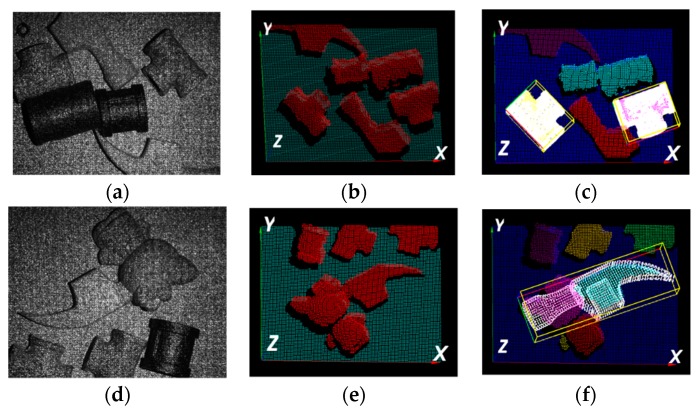
Example of measurement using the NTU-3D Probe: (**a**,**d)** show acquired speckle images; (**b**,**e)** show acquired 3D point clouds; (**c**,**f)** show object recognition and localization results.

**Figure 23 sensors-19-00764-f023:**
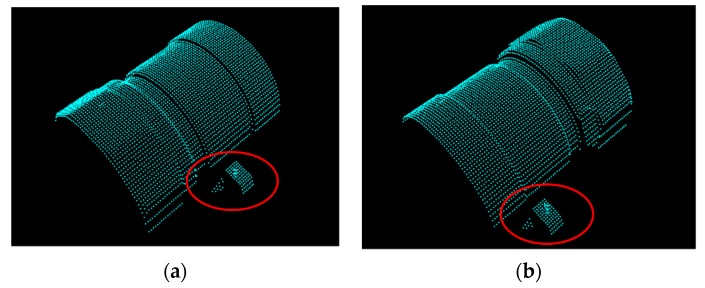
Two views of finefood model in ITRI database. (**a**) Scene no. 19 and (**b**) Scene no. 28.

**Table 1 sensors-19-00764-t001:** Dimensions of objects in ITRI database.

Model		Dimensions (mm)		
Name	3-D representation	Length	Width	Height
Brazo control	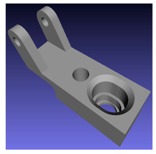	86	53	30
Cylinder	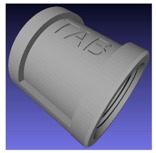	35	35	35
Finefood	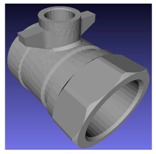	41	40	33
Hammerhead	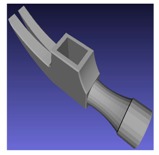	110	35.5	22
Socket	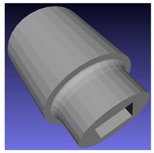	45	35	35
Wrench	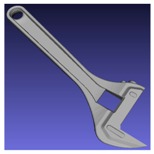	154	65	13.2

**Table 2 sensors-19-00764-t002:** A comparison of performances obtained by the proposed method and existing methods on simulated data.

Models	Method	Translation Error (mm)	Rotation Error (10^−3^ rad)	Time (s)
Fine food	Graph-based	0.869	0.103	2.537
	Feature-based	0.412	0.072	1.453
	View-based	0.036	0.087	7.521
	Proposed	0.019	0.068	0.160
Cylinder	Graph-based	0.435	0.145	4.896
	Feature-based	0.132	0.768	2.902
	View-based	0.038	0.108	8.247
	Proposed	0.022	0.061	0.182
Wrench	Graph-based	0.213	0.315	4.184
	Feature-based	0.896	0.979	3.262
	View-based	0.078	0.113	9.163
	Proposed	0.017	0.081	0.150

**Table 3 sensors-19-00764-t003:** Matching performance for objects in Figure 21.

Case study	Object	Matching Score (%)	Mean Deviation *µ* (mm)	Standard Deviation *σ* (mm)
1	Red	89.74	0.320	0.392
	Magenta	90.12	0.389	0.426
	Cyan	85.55	0.469	0.484
2	Red	98.77	0.249	0.313
	Yellow	99.23	0.187	0.242
	Magenta	97.55	0.247	0.316
	Cyan	92.88	0.351	0.431
	Purple	91.74	0.329	0.394
	Gold	98.55	0.231	0.305
	Spring green	94.57	0.313	0.372
3	Red	98.31	0.266	0.264
	Yellow	99.26	0.180	0.168
Average:		94.69	0.294	0.342

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
