# Peer review of "A Novel Surface Descriptor for Automated 3-D Object Recognition and Localization"

_sensors, 2019, doi:10.3390/s19040764_

Round 1

Reviewer 1 Report

My comments and questions are given in the attached file.

Author Response

Please refer to the response to the reviewer's comment.

Reviewer 2 Report

Dear Authors:

The research article is well written. Object detection and localization from 3D point clouds is an interesting research topic and finds a great application in automated manufacturing and assembly systems. There are couple of 3D descriptors available for the task and the idea presented in the paper looks novel. My comments are given below. First the specific comments are written and then generalized comments.

Specific comments:

- Lines 114 to 117: The section numbers are all zero. They need to be updated as per the correct section numbers.

-Line 238: Figure 9: Which bars are for region-based descriptors and which are for simplified descriptors ? The legends in the bar chart can be improved.

- Line 262: Error! Reference source not found.

- Section 2.4: How many views of the model are used in the template for each point cloud ?

- Line 307: What was the preset threshold ?

- Figure 22: in (c) and (f), are only two point clouds recognized ? Because the bounding box  is shown only for them. What about the other objects in the scene?

Line 414: Table 3 caption: Error! Reference source not found.

Line 433: Error! Reference source not found.

Generalized comments:

- What were the sizes of point clouds ?

- What 3D scanner/sensor was used to generate point clouds ?

- What software/libraries were used for point cloud processing ?

- It would be nice if you can break down the total time into steps. For example how much time from the total time was taken for segmentation, descriptor computation, descriptor matching and refining results ?

- For bin picking application, will the developed algorithm work if all the objects in the bin are of the same type? How would it handle the task of recognition and localization? How would it decide which object to select first ?

- Could you share your database with the research community by uploading it online ? Other researchers can use it and compare the results with the developed algorithms.

- For matching of feature vectors, could you use machine learning (SVM or similar) as  the future work ?

All the best!

Author Response

(The authors gave the same response as above.)

Round 2

Reviewer 1 Report

Thank you for addressing all my comments. My questions have been explained in detail. I believe that the manuscript has been significantly improved.